# Protein Expression Profile in Rat Silicosis Model Reveals Upregulation of PTPN2 and Its Inhibitory Effect on Epithelial-Mesenchymal Transition by Dephosphorylation of STAT3

**DOI:** 10.3390/ijms21041189

**Published:** 2020-02-11

**Authors:** Ying Zhu, Jingxin Yao, Yuxia Duan, Hong Xu, Qiyun Cheng, Xuemin Gao, Shumin Li, Fang Yang, Heliang Liu, Juxiang Yuan

**Affiliations:** School of Public Health, North China University of Science and Technology, The Hebei Key Laboratory for Organ Fibrosis Research, No. 21 Bohai Road, Caofeidian county, Tangshan 063210, China; zhuying@ncst.edu.cn (Y.Z.); yjx9712@163.com (J.Y.); duanyuxia1227@163.com (Y.D.); xuhong@ncst.edu.cn (H.X.); qiyun0710@163.com (Q.C.); gaoxm0623@163.com (X.G.); lishumin@ncst.edu.cn (S.L.); fangyang@ncst.edu.cn (F.Y.); liuheliang@ncst.edu.cn (H.L.)

**Keywords:** proteomics, biomarker, silicosis, PTPN2, EMT, STAT3

## Abstract

Silicosis is a chronic occupational lung disease caused by long-term inhalation of crystalline silica particulates. We created a rat model that closely approximates the exposure and development of silicosis in humans. Isobaric tags for relative and absolute quantitation (iTRAQ) technologies we used to identify proteins differentially expressed in activated rat lung tissue. We constructed three lentiviral knockdown vectors and an overexpression vector for the protein tyrosine phosphatase non-receptor type 2 (*PTPN2*) gene to achieve stable long-term expression. A total of 471 proteins were differentially expressed in the silicosis group compared with controls. Twenty upregulated, and eight downregulated proteins exhibited a ≥1.5-fold change relative to controls. We next found that the PTPN2, Factor B, and VRK1 concentrations in silicotic rats silicosis and SiO_2_-stimulated MLE-12 cells were significantly higher than control groups. More importantly, we found that overexpression of *PTPN2* simultaneously decreased the expression of phospho–signal transducer and activator of transcription 3 (p-STAT3) and Vimentin, while increasing E-cadherin expression. The opposite pattern was observed for *PTPN2*-gene silencing. We identified three proteins with substantially enhanced expression in silicosis. Our study also showed that PTPN2 can inhibit epithelial-mesenchymal transition by dephosphorylating STAT3 in silicosis fibrosis.

## 1. Introduction

Silicosis is a highly prevalent occupational condition that is caused by chronic inhalation of crystalline silica particulates (aerodynamic diameter < 5 μm) into the distal air spaces of the lung. In developing countries, such as China, the control and prevention of silicosis remain unachievable in a short time, requiring significant progress going forward [1]. Persistent inflammation and pulmonary fibrosis are the most common histological changes during silicosis [2]. Accumulating evidence indicates that silica particles activate macrophages and epithelial cells, causing them to release copious amounts of oxidants and cytokines, which leads to fibroblast proliferation, epithelial-mesenchymal transition (EMT), deposition of the extracellular matrix, and ultimately, fibrosis (silicosis) [3,4,5]. However, the complex biological and molecular mechanisms underlying silicosis have not yet been fully elucidated [3].

With the rapid development of proteomics technology and the application of high resolution mass spectrometry, isobaric tags for relative and absolute quantification (iTRAQ) is currently the most widely used technology in quantitative proteomics [6]. The core principle of iTRAQ is polypeptide labeling and quantification, which resolves quantitative changes in protein intensities with high sensitivity, accuracy, and reproducibility [7]; it is widely used in protein biomarker validation to support preclinical and clinical studies [8].In a previous iTRAQ technology study of silicosis, researchers employed SiO_2_ to stimulate raw264.7 cells [9], and TGF-β to stimulate lung fibroblasts [10]. However, nobody employed iTRAQ technology to screen the differential proteins in lung tissue of rats exposed to dynamically silica.

PTPN2 is an intracytoplasmic tyrosine phosphatase with a nuclear localization sequence. PTPN2 can dephosphorylate many protein tyrosine kinase targets, such as the insulin receptor, epidermal growth factor receptor, Src family kinases, Janus kinase, andthe signal transducer and activator of transcription (STAT) family members [11]. Recent studies have found that an abnormally activated STAT3 signaling pathway participates in the development of epithelial-mesenchymal transition (EMT). After the dephosphorylation of STAT3, EMT is inhibited [12]. EMT was closely related to silicosis; therefore, PTPN2 might be the key protein in the process of silicosis.

In this study, we created a rat model that closely approximates the exposure and development of silicosis in humans. iTRAQ coupled with liquid chromatography mass spectrometry (LC/MS) technology, were usedto identify differential expression in activated rat lung tissue. This information was used to assess whether similar markers were upregulated in SiO_2_-stimulated murine lung epithelial (MLE)-12 cells, and the mechanism of PTPN2 inhibiting silicosis fibrosis was studied.

## 2. Results

### 2.1. Protein Expression Profiles Undersilica Exposure

Three technical replicates were analyzed to demonstrate the reproducibility of the experimental results and to perform complementation. Proteins extracted from lung tissue from three biological replicates in each rat model group were pooled for the iTRAQ analysis. The present study identified 471 proteins that were differentially expressed in the silicosis group compared with controls, including 252 with increased and 219 with decreased proteins levels (fold change >1.2 and *p* < 0.05), respectively (Figure 1A and Appendix A). Moreover, 20 increased proteins and 8 decreased proteins exhibited a ≥ 1.5 fold-change (Table 1) relative to the controls. Clustering and heatmap analyses were carried out based on the data (Figure 1B).

### 2.2. Gene Ontology, KEGG Pathway Analysis and Protein-Protein Network Analysis of Differentially Abundant Proteins

Gene ontology (GO) terms were further assigned to differentially expressed proteins according to their cellular components, molecular functions and biological processes (Figure 1C–E and Appendix A). From the cellular component perspective, it can be noticed that the top two differentially expressed proteins were enriched in the integral components of membrane and extracellular exosome. The G-protein-coupled receptor activity and olfactory receptor activity were the top two significantly enriched terms. From the perspective of biological processes, the two most enriched terms were G-protein coupled receptor signaling pathway and regulation of DNA-templated transcription.

We used KOBAS software [13] to test the statistical enrichment of DEGs in KEGG pathways. In this study, 471 differentially expressed proteins involve 217 pathways (Appendix A). Figure 1F shows the results of pathways enrichment analysis, clearly displaying that olfactory transduction pathways constituted the top enriched term. Fourteen differentially expressed proteins identified in our study participate in the olfactory transduction pathways. Moreover, it is worth noting that spliceosome, RNA transport, biosynthesis of antibiotics, protein processing in the endoplasmic reticulum, and Huntington’s disease appeared more often than other pathways.

Interaction between proteins within cells can reveal the function of proteins at the molecular level. This analysis focuses on the construction of networks of differentially expressed proteins, that are of interest to researchers, in order to identify the relationships between these differentially expressed proteins and their functional groups. The protein-protein interaction (PPI) network is shown in Figure 1G.

### 2.3. Validation of iTRAQ-Screened Proteins Exhibiting Increased Expression Levels in the Different Experimental Groups of Silicotic Rats

The progression of silicosis was observed by H&E staining. As shown in Figure 2, the alveolar walls in the control group were thin and clear. In the silicosis model after four weeks, the infiltration of macrophages was obvious in the alveoli and lumens, as well as around blood vessels. The silicosis model at 12 weeks exhibited multiple cellular nodules and isolated nodules composed of macrophages. Western blot results demonstrated that silica inhalation elevated the expression levels of α-SMA and Col I.

To determine whether findings in the late stages of disease were representative of changes in early- and middle-stages, we next performed Western blot analysis on control and silicotic rat lung tissues for those proteins whose expression changed most significantly (≥1.5-fold increase). However, our analysis was limited to proteins for which a commercial antibody for Western blot was available; this included PTPN2, VRK1, factor B, C4BPB, ENT3, GPNMB, SPHKAP, PLBL1, and OSBP for protein identification. Based on our preliminary experimental results, we selected 4-, and 12-weekssilicotic rat models as early- and middle-stage disease models. As shown in Figure 2, Western blot analysis revealed a marked increase in the expression of factor B, PTPN2, and VRK1 in the lung tissues of silica exposed rats, including early- and mid-stage diseases. However, there was no significant difference in other proteins.

Our results show that the alveolar walls were wider and showed a disorganized structure. Cellular fibrous nodules formed after 24 weeks of silica exposure. There were fewer E-cad-positive cells in the model group than in the control group at 24 weeks, but the expression levels of α-SMA and Vimentin were strikingly higher in silicotic rats. (Figure 3A) Western blot results demonstrated that silica inhalation elevated the expression levels of factor B, VRK1, α-SMA, Vimetin, and Col I, and the expression of PTPN2 and E-cad was decreased.

### 2.4. Validation of iTRAQ-Screened Proteins Exhibiting Increased Expression Levels in SiO_2_-Stimulated MLE-12 Cells

To determine whether in vivo findings were representative of changes in vitro, we next performed western blot analysis on control and SiO_2_-stimulated MLE-12 cells for those proteins. Our analysis was also limited to proteins in which a commercially available antibody for Western blot was available; this includedPTPN2, VRK1, factor B, C4BPB, ENT3, GPNMB, SPHKAP, PLBL1, and OSBP for protein identification.

As shown in Figure 4, Western blot analysis revealed a marked increase in the expression of PTPN2, VRK1, C4BPB, Factor B, Vimentin, Col I and α-SMA in SiO_2_ stimulated MLE-12 cells, and the expression of E-cad was decreased.

### 2.5. PTPN2 Overexpression Inhibited the SiO_2_-Stimulated Epithelial-Mesenchymal Transition of MLE-12 Cells by Dephosphorylating STAT3

Protein tyrosine phosphatase non-receptor type 2 (PTPN2) is an intracytoplasmic tyrosine phosphatase. We were unable to find any reports linking either PTPN2 to silicosis, and it has been reported that IL-6 family of cytokines, which signal through STAT-3, may contribute to lung fibrosis [14]. However, one of the major mechanisms that regulated STAT3 activation was dephosphorylation of the tyrosine residue essential for its activation by protein tyrosine phosphatases (PTPs) [15]. Based on this, we hypothesized that changes in the expression of PTPN2 can influence fibrotic responses to SiO_2_ in MLE-12 cells. To test this, we used a lentivirus vector to overexpress the *PTPN2* gene in MLE-12 cells. The results of the lentivirus titer are shown in Table 2.

Co-expression of PTPN2 and E-cad were examined by immunofluorescent staining. As shown in Figure 5A, co-expression of PTPN2 and E-cad was significantly higher in groups of the *PTPN2*-overexpressing lentiviral vector (LW1049) compared to control groups (NC-LW299). The overexpression efficacy of the *PTPN2* lentivirus was also determined by analyzing the protein levels, via western blotting, for the NC-LW299 and LW1049 groups. The results show that the expression levels of PTPN2 were markedly higher in the overexpression group compared with those of the control group. This demonstrated that the lentiviral vectors were effective for *PTPN2* overexpression. Western blotting showed that p-STAT3 was significantly downregulated and E-cadherin (E-cad) was upregulated in the *PTPN2*-overexpression group (Figure 5B).

We then stimulated NC-LW299 and LW1049 cells with SiO_2_. As shown in Figure 6, overexpression of PTPN2 markedly reduced levels of Vimentin, p-STAT3, and collagen I (COL I) and increased E-cad expression in SiO_2_-treated MLE-12 cells. Based on these results, we speculated that PTPN2 overexpression inhibited the silica-stimulated epithelial-mesenchymal transition (EMT) of MLE-12 cells by dephosphorylating STAT3.

### 2.6. PTPN2-Gene Silencing Promoted the SiO_2_-Stimulated EMT of MLE-12 Cells by Phosphorylating STAT3

To confirm the effects of PTPN2 overexpression, three target shRNAs against the mouse PTPN2 gene were constructed for RNAi and transfected into the MLE-12 cells. The lentivirus titer results are shown in Table 3. As shown in Figure 7A, the infection efficiency of the *PTPN2*-gene silencing lentiviral (LW265, LW266, and LW267) and negative control (NC-LW198) vectors in MLE-12 cells was >80%. Because the quantity of gene silencing varied with different shRNAs, we only utilized the shRNA demonstrating the most effective silencing, which was *PTPN2* shRNA03, as shown in Figure 7B. The efficacy of the *PTPN2*-silencing lentivirus in MLE-12 cells at the protein level was determined by western blotting for the NC-LW198 and LW267 groups. The results show that the expression levels of PTPN2 in LW267 were markedly reduced compared with those in the control group. This demonstrates that the lentiviral vectors were effective for *PTPN2* gene silencing. The western blot shows that p-STAT3 was significantly upregulated and E-cad was downregulated in the LW267 group (Figure 7C).

As shown in Figure 8, when we stimulated NC-LW198 and LW267 cells with SiO_2_, the opposite effect to PTPN2 overexpression was seen.

## 3. Discussion

Proteomics involves the large-scale study of proteins and employs techniques to identify complete protein complements of the expressed genome, providing a macroscopic view of what is expressed and present under different growth conditions. With iTRAQ, the protein expression levels can be compared simultaneously [16]. iTRAQ technology has been applied to muchresearch: To identify prognostic biomarkers for gastric cancer [17]; to reveal potential novel biomarkers for the early diagnosis of acute myocardial infarction within three hours [18]; to identified SAA and ACTB as potential biomarkers in patients with severe hand, foot, and mouth disease (HFMD) [19], to discover glioblastoma serum markers [20], and so on.

In a previous iTRAQ technology study of silicosis, nobody to screen the differential protein of silicosis fibrosis in lung tissue of rats exposed to dynamically silica based on iTRAQ technology. Many proteins and small molecules are involved in the development of silicosis fibrosis, and they are the first to change in affected lung tissue. In this study, we created a rat model that closely approximates the development of silicosis in humans, following exposure to crystalline silica. iTRAQ coupled with LC/MS technology, was used to identify activated rat lung tissue and to assess whether the differentially expressed proteins are appropriate factors for the early diagnosis of silicosis. From our results, we identified 471 proteins that were differentially expressed relative to the controls. Importantly, many of these proteins were novel markers that had not previously been linked to pulmonary fibrosis.

This study verified that Protein PTPN2, factor B, and vaccinia-related kinase 1 (VRK1) were significantly upregulated in rat silicosis and SiO_2_-stimulated MLE-12. However, no previous literature has reported that these three proteins were related to pulmonary fibrosis. Factor B is a single-chain glycoprotein composed of 733 amino acid residues. It is mainly synthesized by the liver and macrophages [21,22]. Studies have shown that when macrophages receive stimuli, such as TGF-β1 or LPS, factor B is synthesized and released by macrophages [23,24,25]. A large number of studies have shown that both low and high expression levels of VRK1 have significant regulatory effects on the proliferation and survival of cell lines in normal and malignant tissues [25,26,27]. Studies have shown that VRK1 overexpression reduces both single-cell and sheet migration, these changes are accompanied by a downregulation of the EMT transcriptional repressors snail, slug, and twist1, and an upregulation of the epithelial cell-cell adhesion molecules E-cadherin and claudin-1 [28].

PTPN2 can dephosphorylate activator of transcription (STAT) family members. Studies have shown that STAT family members play an important role in the occurrence and development of pulmonary fibrosis; STAT-3 signaling regulates fibroblasts during the development of pulmonary fibrosis. Furthermore, STAT-3 is activated following injury to type II alveolar epithelial cells, promoting the profibrotic milieu that leads to the accumulation of fibroblasts and myofibroblasts. STAT-3 is also active in lung fibroblasts and is likely to contribute to the fibrotic process through the differentiation of fibroblasts into myofibroblasts [14]. Because the lentivirus vectors provide efficient gene delivery, cause minimal immune reactions in vivo, and can be integrated into the non-dividing cell genome to achieve stable long-term expression [29], we constructed three lentiviral shRNA knockdown vectors and an overexpression vector for the *PTPN2* gene. Our study showed that PTPN2 concentrations were significantly higher in rat silicosis and SiO_2_-stimulated MLE-12 cells than in the control groups. More importantly, we found that overexpression of PTPN2 simultaneously reduced the expression of p-STAT3 and Vimentin, while increasing the expression of E-cadherin. In addition, the effect of *PTPN2* gene silencing was the opposite of that of its overexpression. These results indicate that PTPN2 can inhibit EMT by dephosphorylating STAT3 in silicosis fibrosis.

In conclusion, we found three proteins, PTPN2, factor B, and VRK1, to be significantly upregulated in silicosis. PTPN2 can inhibit EMT by dephosphorylating STAT3 in silicosis fibrosis. Further studies are necessary to uncover whether the individual proteins are innocuous markers of pulmonary fibrosis.

## 4. Materials and Methods

### 4.1. Animals

Specific pathogen-free (SPF) male Wistar rats (three weeks old; weight 180 ± 10 g) were purchased from Liao Ning Changsheng Technology Co. Ltd. (SCXK 2015–0001; Liaoning, China). The rats were housed in the SPF-class laboratory animal room of North China University of Science and Technology, which is consistent with the National Institutes of Health (NIH) guidelines for the care and use of laboratory animals. Animal studies were performed with a protocol approved by the Institutional Animal Care and Use Committee of the North China University of Science and Technology, Tangshan, China (No. LX201868) on April 15, 2018, all animals were carefully handled in compliance with the guidelines.

### 4.2. Rat Models

The rat model was established using a HOPE MED 8050 exposure control apparatus (HOPE Industry and Trade Co. Ltd., Tianjin, China), a noninvasive instrument consisting of a cabinet in which specific concentrations of dust can be set for inhalation by animals [30,31]. The ensuant rat model closely approximates the exposure and development of silicosis in humans.

SiO_2_ (80% with a particle diameter between 1 and 5 μm, s5631; Sigma-Aldrich, St. Louis, MO, USA) was ground and baked at 180 °C for six hours for molding purposes. The settings of the exposure control apparatus were as follows: Pressure −50 to +50 Pa, oxygen concentration 20%, cabinet temperature 20–25 °C, humidity 70–75%, and mass concentration of SiO_2_ 50±10 μg/m^3^. Each animal inhaled for three hours per day.

Rats were randomly allocated into six experimental groups (n = 10) described as follows: Groups 1 to 3 were control groups inhalation of pure air without SiO_2_ for four weeks, 12weeks and 24 weeks, respectively; 4 and 5 were silicotic groups inhalation of SiO_2_ for 4 and 12weeks, respectively; and group 6 inhalation of SiO_2_ for 16 weeks followed by rearing under normal conditions to 24 weeks. Rats were sacrificed by laparotomy following anesthesia. Lungs were carefully removed. Portions of tissue samples were immediately frozen in liquid nitrogen for western blot analysis and fixed in 4% formaldehyde for morphological detection.

### 4.3. Protein Extraction, Digestion, and iTRAQ Labeling

Based on the previous experimental results of our research group [30], the control group at 24 weeks and the silicotic group at 24 weeks were chosen for further iTRAQ studies. Proteins from three biological replicates in each group were extracted using the phenol extraction method [32]. Each sample was extracted three times. The Bradford method [33] was used to measure the concentration of extracted proteins. Approximately 100 μg of pooled protein samples was incorporated into 4 μL reducing reagent (AB Sciex, PN: 4381664; Framingham, MA, USA), and incubated at 60 °C for 1 h; 2 μL cysteine-blocking reagent (AB Sciex, PN: 4381664; Framingham, MA, USA) was added, and the sample pool was maintained at 20 °C for 10 min then subjected to centrifugation in a 10 K ultrafiltration tube (Sartorious, PN: VN01H02; Göttingen, Germany).The samples were centrifuged at 12,000× *g* for 20 min, after which 100 μL dissolution buffer (AB Sciex, PN: 4381664; Framingham, MA, USA) was added followed by centrifugation for another 20 min. Then 2 μg trypsin (AB Sciex, PN: 4370285 and 4352157; Framingham, MA, USA) in dissolution buffer was added to each ultrafiltration tube. The samples were incubated overnight at 37 °C. The resulting peptides were collected by centrifugation. The filters were rinsed with 50 μL dissolution buffer and centrifuged again. Finally, approximately 100 μL of peptides from each treatment group was collected and labled with iTRAQ reagents following the manufacturer protocol (Applied Biosystems PN:4390812, Foster, CA, USA). The control group was labeled with tags 113, 114, and 115 and the silicotic groups with tags 116, 117, and 118.

### 4.4. Separation of Peptides and LC Mass Spectrometric Analysis

Prior to mass spectrometric analysis, peptides were purified to remove excess labeling reagent by high-performance liquid chromatography (HPLC). The labeled samples were dried then diluted with 100 μL cation exchange binding buffer A (98% ddH_2_O, 2% ACN, pH 10.0). Gradient elution on a 4.6 × 250 mm Durashell-C18 column (5 m, 100 A; Poly LC, DC952505-0; Agela, Torrance, CA, USA) was performed to separate peptides at a flow rate of 0.7 mL/min with elution buffer B (2% ddH_2_O, 98% ACN, pH 10.0).

Eluted peptides were collected and red is solved with 20 μL solvent (2% methanol, 0.1% formic acid), then centrifuged at 12,000× *g* for 10 min; 10 μL supernatant was collected for each sample. MS analysis was performed on a Thermo Scientific™ Q-Exactive™ mass spectrometer (Thermo Fisher Scientific, Waltham, MA, USA). Protein identification of the peptide molecules (two technical replicates for each biological replicate) was conducted using Proteome Discoverer™ 1.4 software (Thermo Fisher Scientific, Waltham, MA, USA).

### 4.5. Bioinformatic Analysis

Functional annotation of the identified proteins was conducted via gene ontology (GO) analysis of biological processes, molecular functions and cellular components. The Kyoto Encyclopedia of Genes and Genomes database (KEGG; http://www.genome.jp/kegg/) was used to identify the majority of the important proteins involved in metabolic and signal transduction pathways. *P* values less than 0.05 were considered statistically significant using a two-tailed Fisher’s exact test. A hierarchical clustering (HCL) analysis of the quantitative data in the four comparative groups was performed using Cluster 3.0 software (Stanford University, Stanford, CA, USA) and visualized using Java Treeview software [34]. Analysis of the protein-protein interaction (PPI) network was performed using the STRING online database (https://string-db.org/) [35].

### 4.6. Cell Culture and Treatment

The mouse lung epithelial 12(MLE-12) cell line were purchased from the Chinese Academy of Sciences cell library (TCHu150, Shanghai, China). Cells were maintained at 37 °C under a humidified atmosphere of 5% CO_2_. MLE-12 cells were cultured in DMEM/F-1250/50 1× with L-glutamine and 15 mM HEPES medium (10-092-CV, CORNING, USA) containing 10% FBS (50216, Bovogen, Melbourne, Australia) and 1% Penicillin/Streptomycin (P170410, PAN-Biotech, Aidenbach, Germany). After serum starvation for 24 h, MLE-12 cells were treated with SiO_2_ (50 μg/mL) for 24 h in serum-free medium.

### 4.7. Design and Construction of Lentivirus Vector

#### 4.7.1. Construction of Plasmid Vector for PTPN2 Gene Knockdown Lentivirus

Three target shRNAs against the Mouse *PTPN2* gene (GeneBank accession number NM_001127177.1) for RNAi were designed using an internet application system (Invitrogen, Carlsbad, CA, USA). A shRNA which had no significant homology to any known mouse genes was used as a negative control [36]. The detailed sequences are shown in Table 3.

Three fragments of shRNA were inserted into pHS-ASR-HS-LV008 (pLV-HU6-BsaI-hEF1α-EGFP-2A-Puro) as pHS-ACR/ASR-LW265/266/267/198 (pLV-HU6-*PTPN2*shRNA01/*PTPN2*shRNA02/*PTPN2* shRNA03/neg-control shRNA-hEF1α-EGFP-2A-Puro) with BsaI and T4 Ligase. pHS-ASR-HS-LV008 (pLV-HU6-BsaI-hEF1α-EGFP-2A-Puro) was purchased from/constructed by Beijing Syngentech Co., LTD.

#### 4.7.2. Construction of Plasmid Vector for PTPN2 Gene Overexpress Lentivirus

Based on the website database of National Center of Biotechnology Information (NCBI, USA), the CDS sequence of the *PTPN2* gene (CCDS50311.1) was obtained. Synthetic *PTPN2* CDS was inserted into pHS-BVC-LW299 (pLV-hefla–EGFP–T2A-puro-WPRE-CMV-3Xflag) as pHS-AVC-LW1049 (pLV-hefla-EGFP-T2A-puro-WPRE-CMV-ptpn2-3Xflag) with Enzyme1, Enzyme2, and T4 Ligase. The control plasmid was the skeleton plasmid pHS-BVC-LW299 (pLV-hefla–EGFP–T2A-puro-WPRE-CMV-3Xflag) which was purchased from/constructed by Beijing Syngentech Co., LTD.

#### 4.7.3. The Process of Lentivirus Packaging Plasmids

Lentivirus packaging plasmids were transfected into HEK-293FT cell line with EpFect reagent (Beijing Syngentech Co., LTD.). The suspension was harvested 48 h and 72 h after transfection. Purified and concentrated viral particles (2 × 10^8^ TU/mL) were suspended in HBSS and stored at −80 °C. Titer of lentivirus was calculated with qRT-PCR method. Lentiviru virus was purchased from Beijing Syngentech Co., LTD.

### 4.8. Lentiviral Vector Infection

Cells were divided into six groups (and infected with the respective vectors): (1) NC-LW198 (negative control lentiviral vector, pHS-ASR-LW198); (2) LW265 (protein tyrosine phosphatase non-receptor type 2 [*PTPN2*] [3]-gene silencing pHS-ASR-LW265 lentiviral vector); (3) LW266 (*PTPN2*-gene silencing pHS-ASR-LW266 lentiviral vector); (4) LW267 (*PTPN2*-gene silencing pHS-ASR-LW267 lentiviral vector); (5) NC-LW299 (negative control lentiviral vector, pHS-BVC-LW299); and (6) LW1049 (*PTPN2*-gene overexpressing pHS-AVC-LW1049 lentiviral vector).

Cells were cultured in 6-well plates and infected with specific or negative control lentiviral vectors and 8 µg/mL polybrene (Sigma, St. Louis, MO, USA), at the multiplicity of infection (MOI) 20, according to the pre-experimental results. After incubation for 48 h, cells were observed under a fluorescence microscope. Then, the cells were cultured in the resistant medium containing 1 µg/mL puromycin for 10 days (once every two days) for resistance screening. The expression efficiency was analyzed by western blotting.

### 4.9. Western Blot Analysis

Total protein levels were quantified using a Bradford assay (PC0020; Solarbio, Beijing, China) as previously described. Protein lysates (20 μg/lane) or BALF (20 μL/lane) were separated on a 13% gel by SDS-PAGE and were electro-transferred onto polyvinylidene fluoride (PVDF) membranes. The membranes were blocked with 5% non-fat dry milk and incubated with primary antibodies against PTPN2 (DF6629; Affinity Biosciences, Cincinnati, OH, USA), VRK1 (A7745; ABclonal Biotech, Wuhan, China), factor B (GTX103570; GeneTex, San Antonio, Texas, USA), C4BPB (A6362; ABclonal Biotech, Wuhan, China), ENT3 (bs-21250R; Bioss Biotech, Beijing, China), GPNMB (bs-2684R; Bioss Biotech, Beijing, China), SPHKAP (bs-17662R; Bioss Biotech, Beijing, China), PLBD1 (bs-16113R; Bioss Biotech, Beijing, China), OSBP (DF12128; Affinity Biosciences, Cincinnati, OH, USA), COLI (AF0134; Affinity Biosciences, Cincinnati, OH, USA), anti-E Cadherin (ab76055,Abcam, Cambridge, MA, USA), anti-Vimentin(ARG57642,Arigo Biolaboratories, Wuhan, China), and α-SMA (1184-1; Epitomics, Inc., Burlingame, CA, USA) at 4 °C overnight. Membranes were then washed in TBST and incubated with peroxidase-labeled affinity-purified anti-rabbit/mouse IgG (H+L) secondary antibody (074–1506/074–1806; Kirkegaard and Perry Laboratories, Gaithersburg, MD, USA). ECLTM Prime Western Blotting Detection Reagent (RPN2232; GE Healthcare, Hong Kong, China) was used to visualize protein bands under ChemiDoc™ MP (Bio-Rad Laboratories, Inc.). Quantitative analysis of strip by Image-Pro Plus 6.0 software and expressed as a fold-change relative to GAPDH (AC033; ABclonal Biotech, Wuhan, China)

### 4.10. Immunofluorescence Staining

Cell slides were blocked in goat serum for 15 min to block nonspecific antigens and then were incubated with anti-E Cadherin (ab76055, Abcam, Cambridge, MA, USA) and anti-PTPN2 (DF6629, AffinityBiosciences, Cincinnati, OH, USA) or anti-Vimentin (ARG57642, Arigo Biolaboratories, Wuhan, China) overnight at 4 °C. The next day, the slides were washed three times with PBS buffer, followed by either Alexa FluorR488 donkey anti rabbit IgG (H+L) or NovexR dropped on the slides, and incubated in an incubator at 37 °C for 30 min. Followed by counterstained with DAPI (5 mg/mL, Beyotime, Haimen, China), and then evaluated under afluorescence microscope (Nikon, Tokyo, Japan).

### 4.11. Immunohistochemistry Staining

Paraffin sections (4 μm) were deparaffinized to hydrate them for immunostaining. Retrieval of antigens was done by the high pressure method followed by 15 min of incubation with 0.03% H_2_O_2_ to block peroxidase activity. Next, the slides were incubated with anti-α-SMA (ab7817, Abcam), anti-E Cadherin (ab76055, Abcam, Cambridge, MA, USA), and anti-Vimentin (ARG57642, Arigo Biolaboratories, Wuhan, China) and were kept at 4 °C overnight, and then incubated with a secondaryantibody (PV-6000, ZSGB-BIO, Beijing, China) at 37 °C for 30 min. Immunoreactivity was visualized with DAB (ZLI-9018, ZSGB-BIO, Beijing. Counterstaining was performed using hematoxylin. Finally, slides were sealed with neutral balsam. Images were acquired by microscopy (BX53; Olympus, Tokyo, Japan).

### 4.12. Statistical Analysis

Comparisons between two groups were performed using an independent sample *t*-test. Comparisons between multiple independent groups were made using a one-way ANOVA followed by a post-hoc analysis with the Bonferroni test. Values were expressed as means ± SD, with p-values less than 0.05 considered statistically significant.

## Figures and Tables

**Figure 1 ijms-21-01189-f001:**
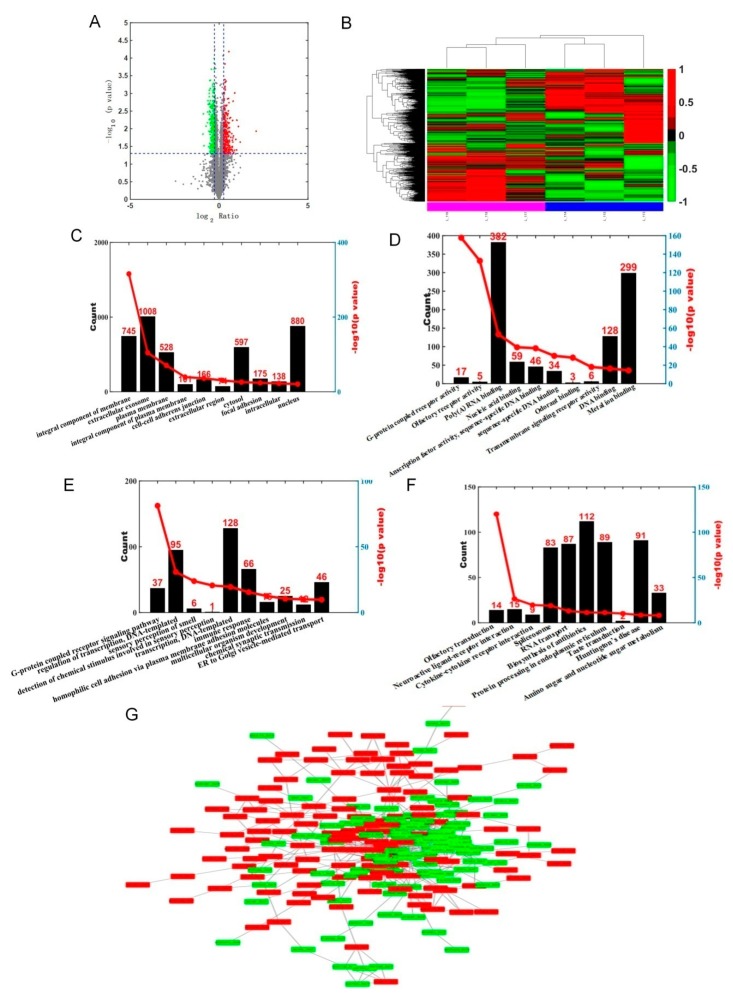
Differentially expressed proteins in silicosis rat lung tissue and bioinformatic analysis. (**A**): Red dots represent upregulated proteins, green dots are downregulated proteins; change was 1.2 fold, *p* value < 0.05 indicated significant difference. (**B**): Each row represents the expression levels of each protein in different samples, and the columns represent the expression levels of all proteins in each sample. The colors indicate the expression levels of the proteins. The larger the expression level, the darker the color (red is upregulated, green is downregulated). The tree chart above shows the clustering analysis results of different samples from different experimental groups, and the tree chart on the left shows the clustering analysis results of different proteins from different samples. (**C**): Cellular component analysis (GO-C). (**D**): Molecular function analysis (GO-F). (**E**): Biological process analysis (GO-P). (**F**): Kyoto Encyclopedia of Genes and Genomes analysis (KEGG). (**G**): Protein-protein interaction (PPI) network.

**Figure 2 ijms-21-01189-f002:**
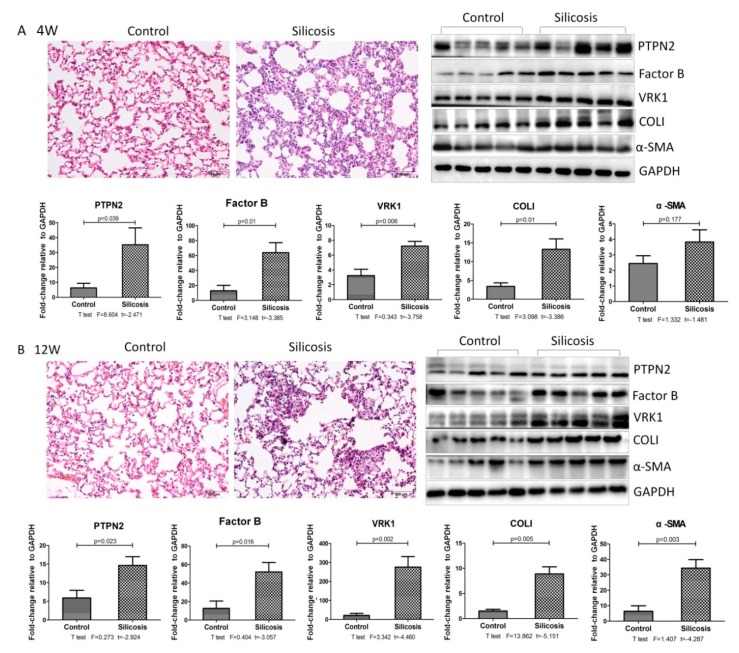
Observation of pathological changes in experimental rat silicosis.(**A**) Lung tissue stained with H&E. Scale bars: 100 μm; Western blot and corresponding densitometry data of the expression levels of PTPN2, factor B, VRK1, Col I, and α-SMA in the lung tissue of the silicosis model after four weeks. (**B**) Lung tissue stained with H&E. Scale bars: 100 μm; Western blot and corresponding densitometry data of the expression levels of PTPN2, factor B, VRK1, Col I, and α-SMA in the lung tissue of the silicosis model after 12 weeks. Relative densities of PTPN2, factor B, VRK1, Col I, and α-SMA were normalized to GAPDH, and the relative expression was expressed as the fold-change of specific bands in the control group. Bar graphs are the mean ± SD of two separate experiments. Statistical analysis was performed using a *t*-test with SPSS20.

**Figure 3 ijms-21-01189-f003:**
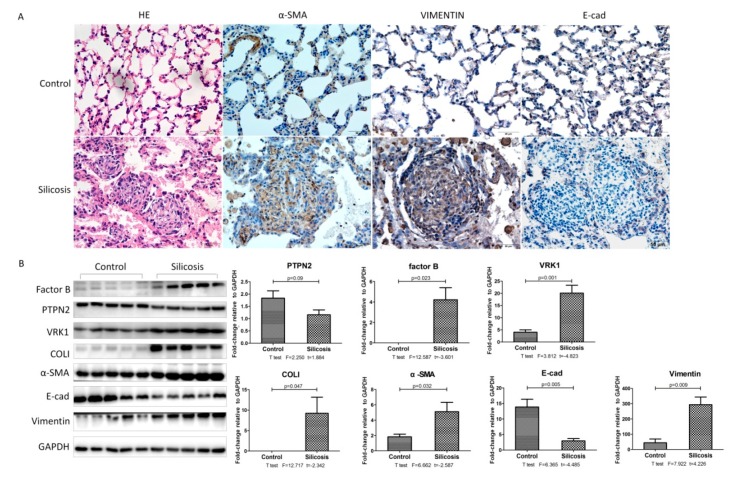
Observation of pathological changes in experimental rat 24Wsilicosis.(**A**) Lung tissue stained with H&E and the expression of α-SMA, Vimentin, and E-cad in the lung tissue of the silicosis model after 24 weeks, measured by immunohistochemistry; Scale bars: 100 μm. (**B**) Western blot and corresponding densitometry data of the expression levels of PTPN2, factor B, VRK1, Col I, and α-SMA in the lung tissue of the silicosis model after 24 weeks. Relative densities of PTPN2, factor B, VRK1, Col I, and α-SMA were normalized to GAPDH, and the relative expression was expressed as the fold-change of specific bands in the control group. Bar graphs are the mean ± SD of two separate experiments. Statistical analysis was performed using a *t*-test with SPSS20.

**Figure 4 ijms-21-01189-f004:**
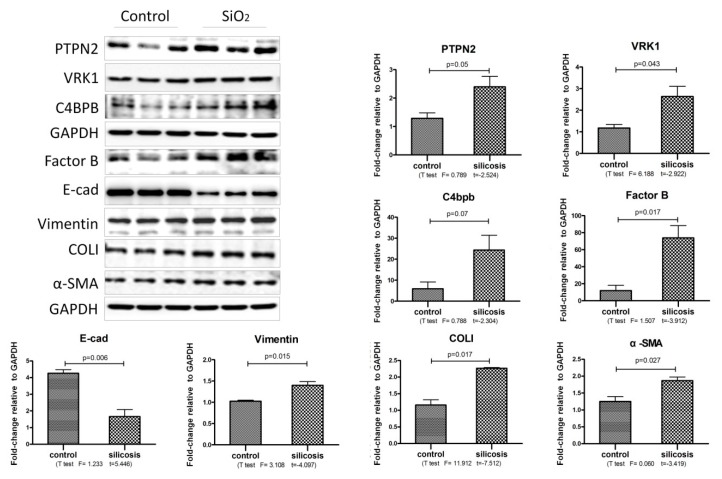
The increased proteins expression in SiO_2_ stimulated MLE-12 cells. Western blot and corresponding densitometry data of the expression levels of PTPN2, factor B, VRK1, C4BPB, E-cad, Vimentin, Col I and α-SMA in MLE-12 cells on SiO_2_ (50 μg/mL) stimulated 24 h. Statistical analysis was performed using a *t*-test with SPSS20.

**Figure 5 ijms-21-01189-f005:**
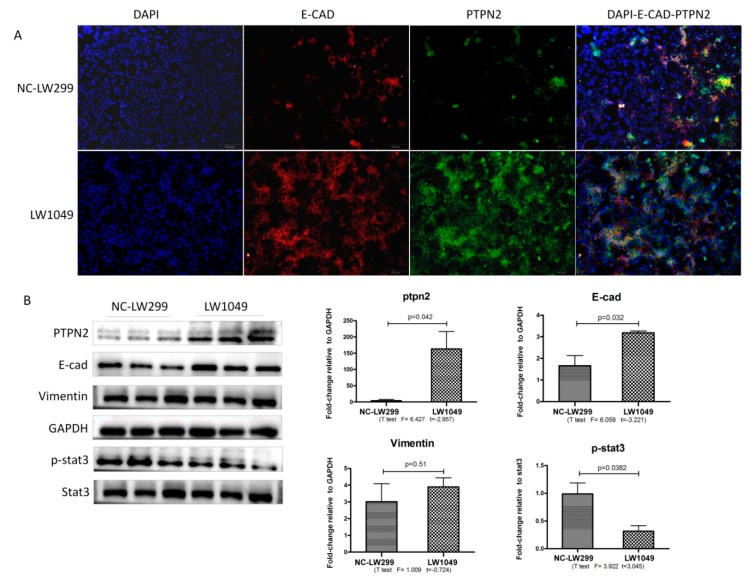
Effect of PTPN2 overexpression on MLE-12 cells. (**A**) Immunofluorescent staining of PTPN2 and E-cad expression for the PTPN2 overexpression (LW1049) and negative control (NC-LW1049) groups; Scale bars: 100 μm. (**B**) Western blot for the expression levels of PTPN2, E-cad, Vimentin, and p-STAT3 in the NC-LW299 and LW1049 cells. Statistical analysis was performed using a *t*-test with SPSS20.

**Figure 6 ijms-21-01189-f006:**
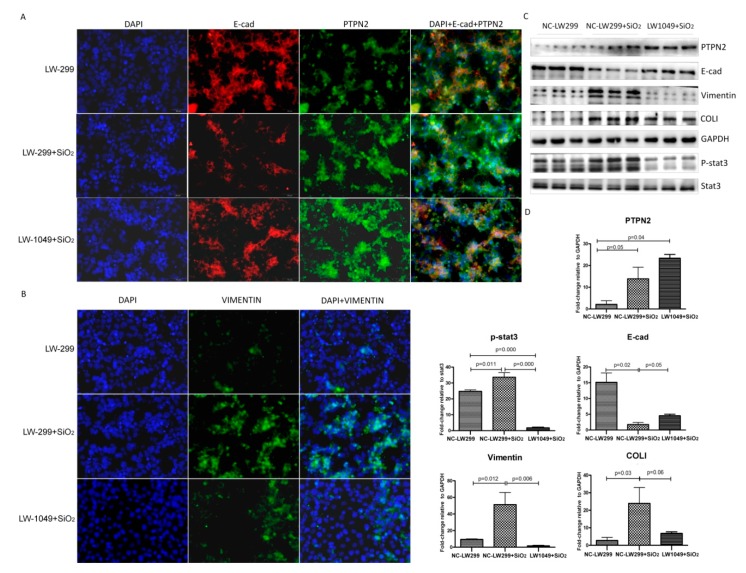
Effect of PTPN2 overexpression on SiO_2_ stimulated MLE-12 cells. (**A**) Immunofluorescent staining for PTPN2 and E-cad expression in NC-LW299 and LW1049 cells with SiO_2_; Scale bars: 50 μm. (**B**) Immunofluorescent staining for Vimentin expression in NC-LW299 and LW1049 cells with SiO_2_; Scale bars: 50 μm. (**C**) Western blot for the expression levels of PTPN2, E-cad, Vimentin, and p-STAT3 in NC-LW299 and LW1049 cells with SiO_2_. (**D**) The corresponding densitometry data for the expression levels of PTPN2, E-cad, Vimentin, and p-STAT3 in NC-LW299 and LW1049 cells with SiO_2_. Relative densities of PTPN2, E-cad, and Vimentin were normalized to GAPDH, and the relative density of p-STAT3 was normalized to STAT3. The relative expressions were expressed as the fold-change of specific bands in the negative control group. Bar graphs are the mean ± SD of two separate experiments, each performed in triplicate. Statistical analysis was performed using a *t*-test with SPSS20.

**Figure 7 ijms-21-01189-f007:**
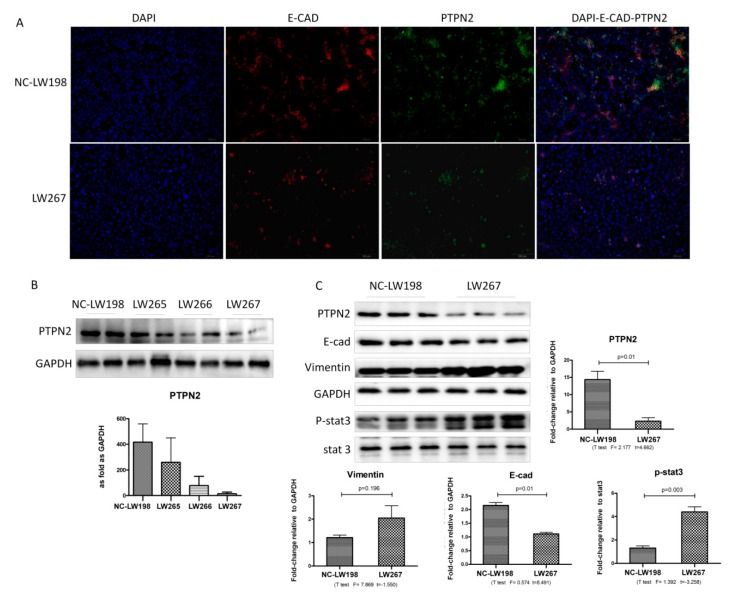
Effect of PTPN2 gene silencing on MLE-12 cells. (**A**) Immunofluorescent staining of PTPN2 and E-cad expression for the PTPN2 gene silencing (LW267) and negative control (NC-LW198) groups; Scale bars: 100 μm. (**B**) Effectiveness of different shRNA probes in the expression of PTPN2. (**C**) Western blot and corresponding densitometry data of the expression levels of PTPN2, E-cad, Vimentin, and p-STAT3 in the NC-LW198 and LW267 cells. The relative expressions were expressed as the fold-change of specific bands in the negative control group. Bar graphs are the mean ± SD of two separate experiments. Statistical analysis was performed using a *t*-test with SPSS20.

**Figure 8 ijms-21-01189-f008:**
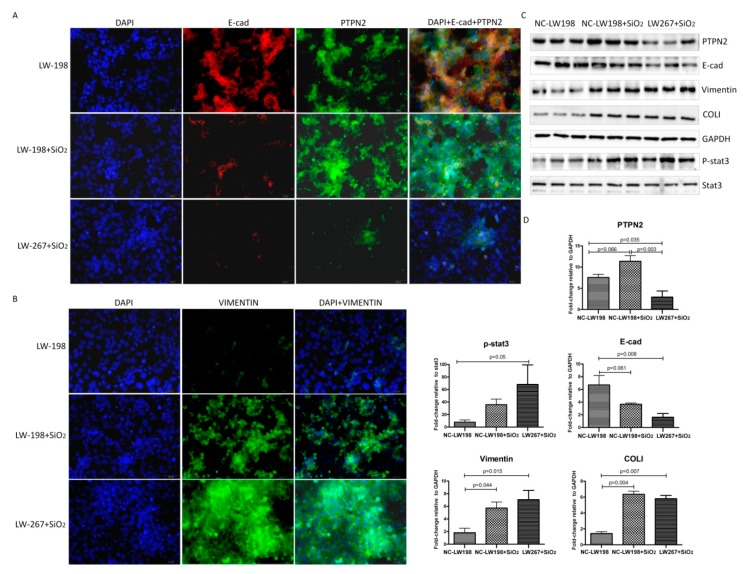
Effect of PTPN2 gene silencing on SiO_2_ stimulated MLE-12 cells. (**A**) Immunofluorescent staining for PTPN2 and E-cad expression in NC-LW198 and LW267 cells with SiO_2_; Scale bars: 50 μm. (**B**) Immunofluorescent staining for Vimentin expression in NC-LW198 and LW267 cells with SiO_2_; Scale bars: 50 μm. (**C**) Western blot for the expression levels of PTPN2, E-cad, Vimentin, and p-STAT3 in NC-LW198 and LW267 cells with SiO_2_. (**D**) The corresponding densitometry data for the expression levels of PTPN2, E-cad, Vimentin, and p-STAT3 in NC-LW198 and LW267 cells with SiO_2_. Relative densities of PTPN2, E-cad, and Vimentin were normalized to GAPDH, and the relative density of p-STAT3 was normalized to STAT3. The relative expressions were expressed as the fold-change of specific bands in the negative control group. Bar graphs are the mean ± SD of two separate experiments, each performed in triplicate. Statistical analysis was performed using a *t*-test with SPSS20.

**Table 1 ijms-21-01189-t001:** Subset of differentially expressed proteins in silicotic rat lung tissue relative to the controls.

No.	Protein Name	Accession Number	Fold Change
1	Acidic mammalian chitinase	[CHIA_RAT]	4.277554
2	Tyrosine-protein phosphatase non-receptor type 2	[PTN2_RAT]	2.292036
3	Protein Vrk1	[VRK1_RAT]	2.126730
4	B-factor, properdin	[Q6MG74_RAT]	1.970988
5	Complement component 4, gene 2	[Q6MG90_RAT]	1.822588
6	Cd68 molecule	[Q4FZY1_RAT]	1.773507
7	Protein Spcs1	[D3ZFK5_RAT]	1.749611
8	Equilibrative nucleoside transporter 3	[ENT3_RAT]	1.731603
9	BPI fold-containing family B member 1	[BPIB1_RAT]	1.716078
10	Monocarboxylate transporter 4	[MOT4_RAT]	1.680098
11	Transmembrane glycoprotein NMB	[GPNMB_RAT]	1.676391
12	A-kinase anchor protein SPHKAP	[SPKAP_RAT]	1.672275
13	Complement C4	[CO4_RAT]	1.652953
14	Olfactory receptor	[D3ZRJ8_RAT]	1.652572
15	Phospholipase B-like 1	[PLBL1_RAT]	1.585530
16	Chitinase 3-like 1 protein (Fragment)	[A4LA56_RAT]	1.558782
17	Cell division cycle protein 123 homolog	[CD123_RAT]	1.546649
18	Protein Stk33	[D4A165_RAT]	1.532094
19	Beta-defensin 4	[DEFB4_RAT]	1.519443
20	UDP-glucuronosyltransferase	[Q6T5E8_RAT]	1.506820
21	Protein Slc16a2	[G3V9C2_RAT]	−1.877088
22	Carboxylic ester hydrolase	[A0A0G2JV37_RAT]	−1.842240
23	Protein Spns2	[D3ZPS4_RAT]	−1.796205
24	PDZ domain-containing protein 2	[F1M785_RAT]	−1.666518
25	Protein Dcp1b	[D3ZZJ7_RAT]	−1.571295
26	CD34 antigen isoform 2	[B1PLB2_RAT]	−1.566583
27	Protein Shroom2	[SHRM2_RAT]	−1.541996
28	Periaxin	[PRAX_RAT]	−1.528930

**Table 2 ijms-21-01189-t002:** Detection results of lentivirus titer.

Vector Number	Lenticirus Titer
pHS-ASR-LW265	1.08 × 10^8^ TU/mL
pHS-ASR-LW266	1.02 × 10^8^ TU/mL
pHS-ASR-LW267	1.37 × 10^8^ TU/mL
pHS-ASR-LW198	1.09 × 10^8^ TU/mL
pHS-AVC-LW1049	1.03 × 10^8^ TU/mL
pHS-BVC-LW299	1.83 × 10^8^ TU/mL

**Table 3 ijms-21-01189-t003:** Gene silencing lentivirus vector information.

Vector Number	Inserted Content	shRNA Sequence
pHS-ASR-LW265	*PTPN2* shRNA01	5′-GCATTCTACGGAAACGTATTC CGAAGAATACGTTTCCGTAGAATGC-3′
pHS-ASR-LW266	*PTPN2* shRNA02	5′-GCATTAATCTCAAGGGTTTGT CGAAACAAACCCTTGAGATTAATGC-3′
pHS-ASR-LW267	*PTPN2* shRNA03	5′-GCAGGTAAGAAGGATGTTTCC CGAAGGAAACATCCTTCTTACCTGC-3′
pHS-ASR-LW198	Neg control shRNA	5′-CCTAAGGTTAAGTCGCCCTCGC CGAAGCGAGGGCGACTTAACCTTAGG-3′

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
