# Peer review of "Protein Expression Profile in Rat Silicosis Model Reveals Upregulation of PTPN2 and Its Inhibitory Effect on Epithelial-Mesenchymal Transition by Dephosphorylation of STAT3"

_ijms, 2020, doi:10.3390/ijms21041189_

Round 1

Reviewer 1 Report

Please see attached Review Report. 

Author Response

February 7, 2020

Dear Reviewer,

Thank you very much for the comments of you concerning our manuscript, “Global protein expression Profile in silicotic rats and PTPN2 exerts

inhibitory effect on epithelial-mesenchymal transition by dephosphorylation

of STAT3” (ID: ijms-707715). Those comments are all valuable and very helpful for revising and improving our paper, and they have provided important guiding significance for our research. We have studied comments carefully and have made corrections that we hope meet with approval. Any revisions were clearly highlighted using the "Track Changes" function in the revised manuscript.

Responses to Reviewer:

Point 1: Title is confusing and need modification as suggested below.

Protein expression profile in rat silicosis model reveals up-regulation of PTPN2 and its inhibitory effect on epithelial-mesenchymal transition by dephosphorylation of STAT3

Line 2: Change Profile to profile

Response 1: Thank you very much for your suggestion. we think the title you suggested is more scientific and rigorous to the manuscript. We have changed the title to“Protein expression profile in rat silicosis model reveals up-regulation of PTPN2 and its inhibitory effect on epithelial-mesenchymal transition by dephosphorylation of STAT3.”

Point 2: Rat Models: This section needs some modifications.

Line 74-78: These should be part of the Results rather than the Methods.

Line 76-78: “Cell fibrous nodules were appeared in silicotic rats at 24 weeks,

accompanied by diffuse interstitial pulmonary fibrosis” - Is this observation based on

your preliminary experiment?

Line 79-82: Allocation of experimental groups is not explained well. Did you chose these six experimental groups based on the preliminary experiment?

Line 87: Based on what preliminary results that 24 weeks’ time point is chosen for

iTRAQ?

Response 2: Thank you very much for your recognition. There are many problems in the description of the time points of rat models.

The time point selection is based on the experiment results of Zhang Lijuan, who is the member of our research group (The 13th reference of our manuscript). Her results on this part are described as follows: “macrophages were visible in the lumen of the alveoli 4 weeks after silica inhalation. Multiple cellular nodules, composed of macrophages, were present in the lung tissue of rats after 8 weeks of silica inhalation. The silica nodules became larger as the duration of silica exposure increased, and multiple fused nodules were present in the lungs after 16 weeks of silica inhalation. Cell fibrous nodules formed after 24 weeks of silica exposure. The nodule area

reached nearly 50% after 32 weeks of silica exposure”.

Based on her results, we chosen the 24 weeks for iTRAQ, and chosen the 4 weeks, 12 weeks, and 24 weeks for Verification of iTRAQ results. As shown on Fig.2 and Fig. 3, the rat silicosis model was successfully established, and the selection of these three time points is appropriate. We revised the manuscript and deleted the irrelevant content (lines 88–101).

Point 3: Line 130: Manufacturer’s name for FBS and antibiotics. Name the antibiotics used.

Response 3: We provide more details about the FBS and antibiotics in the revised manuscript (lines 146-147).

Point 4: Line 175: Primary antibodies names and manufacturers

Response 4: We provide more details about the primary antibodies names and manufacturers in the revised manuscript (lines 195-201).

Point 5: Line 179: Please describe how densitometry of the bands done? Which software used?

Response 5: We provide more details in the revised manuscript (lines 205-208).

Point 6: Line 194: Manufacturer’s name for One step polymer detection system kit

Response 6: We provide more details in the revised manuscript (lines 222-225), and corrected a mistake (line 221).

Point 7: In all the figures, change fold to Fold

Response 7: We changed all the figures, reuploaded, and reinserted into the manuscript.

Point 8: Line 211: Change Fig. 2B to fig. 1B

Line 217: Is not this Fig. 1, C-E? Please change.

Line 226: Change Figure 2F to Figure 1F

Line 235: Change Figure 2G to Figure 1G

Response 8: I'm sorry for the mistakes. We've corrected in the revised manuscript (lines 242-266).

Point 9: Line 255: Along with α-SMA and Col I, VRKI is also elevated. Please modify

Response 9: This information was described in the next paragraph (lines 296-298).

Point 10: Line 306: Modify sentence ----- in MLE-12 cells on SiO2(50μg/ml) stimulated 24 h-----in MLE-12 cells stimulated with SiO2 (50 μg/ml) for 24 h

Response 10: We've modified in the revised manuscript (lines 333-336).

Point 11: Line 353: Change Figure 6A to Figure 7A

Line 357: Change Figure 6B to Figure 7B

Line 362: Figure 7C

Response 11: I'm sorry for the mistakes. We've corrected in the revised manuscript (lines 388-399).

Point 12: Line 395-399: These are repetitive of information that was already mentioned in the Introduction. It’s not necessary to add these in the Discussion.

Response 12:We revised the manuscript and deleted the content (lines 434–438).

Point 13: Please carefully proofread to fix all the punctuation errors.

Response 13: I'm sorry for the mistakes. We've corrected in the revised manuscript.

Point 14: - Please insert space before parenthesis

For example: macrophages[26-27]. These need to be corrected to macrophages [26-27].

Response 14: I'm sorry for the mistakes. We've corrected in the revised manuscript.

Point 15: The introduction can be improved

Response 15: Thank you very much for your suggestion. We think your suggestion is more scientific and rigorous to the manuscript. We added the content about PTPN2 mechanisms and why can be important its study in silicosis to the introduction and deleted the duplicates content in the discussion (lines 55-62, 458–469).

Thank you very much for your kind work and consideration on publication of our

paper. On behalf of my co-authors, we would like to express our great appreciation to

you. We look forward to hearing from you at your earliest convenience.

Thank you and best regards,

Sincerely,

Juxiang Yuan, M.D.

School of Public Health

North China University of Science and Technology

No. 21 Bohai Road, Caofeidian county, Tangshan city, Hebei province 063210, China

Email: yuanjx@ncst.edu.cn

Reviewer 2 Report

This paper reports an iTRAQ-based quantitative proteomics study of the protein expression profile in silicotic rats. In addition, the PTPN2 inhibitory effect on epithelial mesenchymal transition by dephosphorylation was also studied.

In my opinion science is good and results are convincing. The introduction can be improved with some more information about PTPN2 mechanisms and why can be important its study in silicosis. The discussion is well-written and provides interesting and detailed information about pathways by which the differentially expressed proteins may be involved in the silicosis pathologic process. However, despite the interesting approach of this study, an English revision must be performed before this paper can be considered for publication in International Journal of Molecular Sciences. Also, some inaccuracies in the reference of the figures throughout the text have to be corrected.

Please find in attachment the manuscript with some comments and suggestions highlighted in color.

Author Response

February 7, 2020

Dear Reviewer,

Thank you very much for the comments of you concerning our manuscript, “Global protein expression Profile in silicotic rats and PTPN2 exerts

inhibitory effect on epithelial-mesenchymal transition by dephosphorylation of STAT3” (ID: ijms-707715). Those comments are all valuable and very helpful for revising and improving our paper, and they have provided important guiding significance for our research. We have studied comments carefully and have made corrections that we hope meet with approval. Any revisions were clearly highlighted using the "Track Changes" function in the revised manuscript.

Responses to Reviewer:

Point 1: In my opinion science is good and results are convincing. The introduction can be improved with some more information about PTPN2 mechanisms and why can be important its study in silicosis.

Response 1: Thank you very much for your suggestion. We think your suggestion is more scientific and rigorous to the manuscript. We added the content about PTPN2 mechanisms and why can be important its study in silicosis to the introduction and deleted the duplicates content in the discussion (lines 55-62, 458–469).

Point 2: However, despite the interesting approach of this study, an English revision must be performed before this paper can be considered for publication in International Journal of Molecular Sciences.

Response 2: We have revised the whole article. If there is any problem, we will continue to revise it.

Point 3: Also, some inaccuracies in the reference of the figures throughout the text have to be corrected.

Response 3: I'm sorry for the mistakes. We've corrected in the revised manuscript (lines 242-266, 388-399).

Point 4: Please find in attachment the manuscript with some comments and suggestions highlighted in color.

Response 4: We have corrected all the mistakes highlighted in color. We have changed the title to “Protein expression profile in rat silicosis model reveals up-regulation of PTPN2 and its inhibitory effect on epithelial-mesenchymal transition by dephosphorylation of STAT3.” We changed the figure 1 to improve image  resolution, reuploaded, and reinserted into the manuscript. We've modified the sentences in the revised manuscript (lines 333-336). We've improved the sentences of discussion (lines 427-433).

Thank you very much for your kind work and consideration on publication of our

paper. On behalf of my co-authors, we would like to express our great appreciation to you. We look forward to hearing from you at your earliest convenience.

Thank you and best regards,

Sincerely,

Juxiang Yuan, M.D.

School of Public Health

North China University of Science and Technology

No. 21 Bohai Road, Caofeidian county, Tangshan city, Hebei province 063210, China

Email: yuanjx@ncst.edu.cn
